# High-Magnetic-Sensitivity Magnetoelectric Coupling Origins in a Combination of Anisotropy and Exchange Striction

**DOI:** 10.3390/nano12183092

**Published:** 2022-09-06

**Authors:** Zhuo Zeng, Xiong He, Yujie Song, Haoyu Niu, Dequan Jiang, Xiaoxing Zhang, Meng Wei, Youyuan Liang, Hao Huang, Zhongwen Ouyang, Zhenxiang Cheng, Zhengcai Xia

**Affiliations:** 1Wuhan National High Magnetic Field Center & School of Physics, Huazhong University of Science and Technology, Wuhan 430074, China; 2Institute for Superconducting and Electronic Materials, Australia Institute for Innovation Materials, Innovation Campus, University of Wollongong, Squires Way, North Wollongong, NSW 2500, Australia

**Keywords:** multiferroic materials, anisotropy, DyFeO_3_, magnetoelectric coupling, pulsed high magnetic field

## Abstract

Magnetoelectric (ME) coupling is highly desirable for sensors and memory devices. Herein, the polarization (*P*) and magnetization (*M*) of the DyFeO_3_ single crystal were measured in pulsed magnetic fields, in which the ME behavior is modulated by multi-magnetic order parameters and has high magnetic-field sensitivity. Below the ordering temperature of the Dy^3+^-sublattice, when the magnetic field is along the *c*-axis, the *P* (corresponding to a large critical field of 3 T) is generated due to the exchange striction mechanism. Interestingly, when the magnetic field is in the *ab*-plane, ME coupling with smaller critical fields of 0.8 T (*a*-axis) and 0.5 T (*b*-axis) is triggered. We assume that the high magnetic-field sensitivity results from the combination of the magnetic anisotropy of the Dy^3+^ spin and the exchange striction between the Fe^3+^ and Dy^3+^ spins. This work may help to search for single-phase multiferroic materials with high magnetic-field sensitivity.

## 1. Introduction

Multiferroic materials [1,2,3,4,5,6,7], with the coupling of two or more ferroic orders, have been attracting much attention due to their intriguing physics and great application potential. The *Pbnm* structured orthoferrites *R*FeO_3_ (*R* = rare earth element) have great potential value for application as magnetoelectric (ME) devices based on the mutual control of magnetization (*M*) and electric polarization (*P*) [8,9,10,11]. For example, at lower temperatures, the application of a large critical magnetic field along the *c*-axis induces a multiferroic (weakly ferromagnetic of Fe^3+^-sublattice and ferroelectric) state in DyFeO_3_, and the magnetic field induced *P* results from the movement of the Dy^3+^ ions toward the Fe^3+^ ions and backward, corresponding to the exchange striction [12]. The ME coupling and *P* are decided by the spin configurations of both Fe^3+^ and Dy^3+^ ions. In DyFeO_3_, Fe^3+^ ions (*S* = 5/2) exhibit the G*_x_*A*_y_*F*_z_* (magnetic configuration in Bertaut’s notation below its Néel temperature of the Fe^3+^-sublattice *T*_N_(Fe) = 650 K) [13,14,15,16], in which the main component of the magnetic moment of Fe^3+^ ions lies along the *a*-axis, and due to the Dzyaloshinskii–Moriya interaction (DMI), a small fraction of the moment is canted along the *c*-axis, causing weak ferromagnetism (wFM) in the material [17]. With decreasing temperature, the spin-reorientation (Morin) transition occurs at *T_SR_* (in the range of about 35~70 K [18,19,20,21]), where the magnetic configuration of Fe^3+^ ions changes from G*_x_*A*_y_*F*_z_* to A*_x_*G*_y_*C*_z_*, and then the wFM disappears. Below the antiferromagnetic (AFM) ordering temperature of the Dy^3+^-sublattice *T*_N_(Dy) = 4.2 K, the Dy^3+^ magnetic configuration is G*_x_*A*_y_* [22,23] with the Ising axis deviation of about 33° from the *b*-axis. When a magnetic field (higher than about 3 T) is applied along the *c*-axis below *T_SR_*, the spin configuration of the Fe^3+^-sublattice is driven to G*_x_*A*_y_*F*_z_* again [4].

For spin-driven ferroelectricity, there are mainly three types of microscopic mechanism models, i.e., the inverse Dzyaloshinskii–Moriya (IDM) mechanism [24], spin-dependent *p*-*d* hybridization model [25], and exchange striction model [26,27], for explaining ME behaviors. According to these models, the emergence of ferroelectricity is hardly understood only by the local spin arrangement, since the symmetry of the crystal structure needs to be considered. In DyFeO_3_, below *T*_N_(Dy), the *P_c_* (the direction of *P* is parallel to the *c*-axis) can be induced when the magnetic fields (higher than 2.4 T at 3.0 K [4]) are applied along the *c*-axis. However, the ME behaviors in DyFeO_3_, i.e., the *P_c_* is induced with the magnetic fields along other crystal axes, remain to be further understood. Moreover, some abnormal behaviors have been observed. For example, a small *P* drop was observed on the *P*-*H* curve at *B_C_*(Fe) [4]. Abnormal heat transport was measured, and a Fe_III_ state (a metastable phase) was speculated [18]. These abnormal behaviors indicate that there may be complex and delicate magnetic interactions in ME behaviors, such as the competition between the anisotropic energy of the Dy^3+^-sublattice, the coupling energy between the Dy^3+^ and Fe^3+^-sublattices, and Zeeman energy [28].

In DyFeO_3_, when the magnetic field is in the *ab* plane, ME coupling with smaller critical fields of 0.8 T (*a*-axis) and 0.5 T (*b*-axis) is triggered. Inspired by these lower critical magnetic fields, we revisited the structure of DyFeO_3_, in which two AFM sublattices (the Fe^3+^- and Dy^3+^-sublattices) are nesting with each other (see Figure 1a,b). The Fe^3+^-sublattice (blue balls) has strong AFM coupling (G*_x_*A*_y_*) and wFM (F*_z_*), and the Dy^3+^-sublattice (red balls) has weak AFM coupling and strong magnetic anisotropy (the AFM vector is localized in the *ab* plane). Under a lower magnetic field in the easy plane (*ab* plane), the direction of the magnetic anisotropy of the Dy^3+^-sublattice might be disturbed or changed, which leads to the change in exchange striction between the Fe^3+^- and Dy^3+^-sublattices and triggers *P_c_.* Thus, we believe that single-phase materials with nested AFM lattices (as shown in Figure 1c,d) can be designed or found, where the A-sublattice (blue spheres) has strong AFM coupling in the plane and weak FM outside the plane, while the B-sublattice (red spheres) has weak AFM coupling, and the B-site ions have strong magnetic anisotropy. Such AFM systems are expected to achieve highly magnetically sensitive ME coupling induced by in-plane magnetic fields. Although the observed magnetoelectric effects mainly occurred at low temperatures (below *T*_N_(Dy)), which may be difficult to apply directly in the traditional industry, our work deepens the understanding of the ME coupling in DyFeO_3_. On the other hand, the ME systems controlled by a combination of multiple parameters (such as magnetic anisotropy and exchange striction) may have high sensitivity to the external magnetic field.

## 2. Experiment

The DyFeO_3_ polycrystalline sample was synthesized by the solid-phase synthesis method from Dy_2_O_3_ (99.99%) and Fe_2_O_3_ (99.99%), and the DyFeO_3_ single crystal was grown by using a Four Mirror Optical Floating Zone Furnace (Crystal Systems Corp., Salem, MA, USA). The crystal structure and purity of both the DyFeO_3_ powder (crushed single crystal) and the single-crystal samples were measured with an X-ray diffractometer (XRD, X’Pert MPD Powder-DY3734, PANalytical B.V., Almelo, NL) using Cu Ka radiation (λ = 1.5406 Å). The scan ranged from 10 to 90°, the step size was 0.013°, and the scan rate was 0.042°/s. The directions of three principal axes were determined according to the Laue X-ray diffractometer measurement results. The low magnetic-field magnetization was measured by using a superconducting quantum interference device (SQUID VSM, Quantum Design, San Diego, CA, USA). Dynamic behaviors of both the *M* and the *P* were measured under a pulsed high magnetic field at the Wuhan National High Magnetic Field Center. The pulsed-high-magnetic-field *M* was detected by the standard inductive method employing two concentric pick-up coils connected in series with opposite polarity [7]. The electrical polarization *P_c_* measurement schematic is shown in Appendix A. The silver electrodes are evenly distributed on the upper and lower surfaces of the sample. When the magnetic field is applied to the sample, the current signal (corresponding to the change in charge density induced by magnetic fields) in the sample is converted into an electrical signal (V) at the reference resistor (R), and then the V is obtained after being processed by a preamplifier. Finally, the change in the electric polarization induced by the pulsed magnetic fields is obtained.

## 3. Results and Discussion

The XRD pattern of the DyFeO_3_ powder and its fitting by the general structure analysis system (GSAS) are shown in Figure 2a. All the diffraction peaks are well indexed by a distorted orthorhombic structure with *Pbnm*. No impurity peaks are observed within the diffraction resolution, indicating the single-phase nature of the sample. The lattice parameters *a* = 5.3031 Å, *b* = 5.5983 Å, and *c* = 7.6228 Å and the detailed crystal parameters are listed in the Appendix A, which are close to the values in the Inorganic Crystal Structure Database (ICSD 27280). In the diffraction pattern of the single-crystal sample, only the (002), (004), and (006) diffraction peaks are observed, which confirms the high quality and accurate *c*-axis orientation of the DyFeO_3_ single-crystal sample.

The temperature dependence of magnetization measured in various magnetic fields (0.01 T, 0.05 T, 1 T, and 5 T, respectively) is shown in Figure 2b–d. In the lower-temperature region (below ~60 K), the zero-field-cooled (ZFC) and field-cooled (FC) magnetization curves have slight deviation, and the difference is presented in the insets of Figure 2b–d. In the higher-temperature region (above ~60 K), the difference between ZFC and FC becomes indiscernible. When the magnetic field is applied along the *c*-axis (see Figure 2d), there is an obvious transition with the magnetization jumps of ~0.13 µ_B_/f.u. at *T*_SR_(Fe) ~57 K, which is related to the spin-flop transition of the Fe^3+^-sublattice. As the temperature decreases to *T_N_*(Dy), an obvious drop can be observed (see the inset of Figure 2d), which indicates that the Dy^3+^-sublattice undergoes a transition from a paramagnetic state to an AFM state G*_x_*A*_y_*. Since the magnetic moment of the Ising Dy^3+^ ions is localized in the *ab* plane, it is difficult to disturb the magnetic field (0.05 T, along the *c*-axis) or change the direction of the magnetic anisotropy (or the anisotropy energy) of the Dy^3+^ spin. With the magnetic field increasing, *T*_SR_(Fe) moves to the low-temperature region. However, with the magnetic field applied along the *c*-axis, no obvious movement of *T_N_*(Dy) is observed, which confirms the strong magnetic anisotropy and localization in the *ab* plane of the Dy^3+^ spins. The temperature and magnetic field dependence of the transitions are shown in the magnetic phase diagram of Appendix A. In DyFeO_3_, there is the magnetic anisotropy of Dy^3+^ ions, field-induced spin flop of Fe^3+^ ions, temperature-driven spin reorientation of Fe^3+^ ions, AFM interaction between Dy^3+^ and Fe^3+^ ions, and thermal fluctuation [28], and these lead to the complex dependence of both *T*_SR_(Fe) and *T_N_*(Dy) on temperatures and magnetic fields.

The magnetization curves as functions of different magnetic fields along three principal axes are shown in Figure 3. Below *T*_N_(Dy) (taken 2 K as an example), when the magnetic field is applied along the *a*-axis (see Figure 3a)**,** a transition is observed at *B*_C_(Dy) ~0.8 T (labeled with a red arrow), which is attributed to the spin-flop transition of the Dy^3+^-sublattice (as shown in the inserted cartoon). With the magnetic field increasing, the magnetic field drives the spins of the Dy^3+^-sublattice toward the *a*-axis as much as possible, resulting in a sharp increase in *M.* With the magnetic field further increasing, the AFM coupling of Dy^3+^-sublattices may be partially broken. For the magnetic field parallel to the *b*-axis, similarly, a slightly smaller critical magnetic field of 0.5 T can be observed due to the smaller deviation (the angle ~33°) of the Ising vector of the Dy^3+^ spins from the *b*-axis. Above 0.5 T, a saturated magnetic moment of ~8.3 µ_B_/f.u. is obtained, which indicates that the Dy^3+^ moment was almost magnetized to saturation by the magnetic field (as shown in the inserted cartoon in Figure 3b). The saturated magnetization shows that the Ising Dy^3+^ moment is mainly localized in the *ab* plane with the *b*-axis as the easy axis [18]. In the case with the magnetic field along the *c*-axis, a magnetization jump (Δ*M*) of ~0.13 µ_B_/f.u. is observed around 3 T. The value of the Δ*M* is the same as the value of the magnetization jump in the *M*-*T* curves shown in Figure 2d, which confirms that the transition is mainly associated with the spin-flop transition of the Fe^3+^-sublattice; the result is also consistent with previous studies [4]. In Figure 3c, two transitions are observed around 3 T and 3.2 T (labeled as black and red triangles, respectively). The lower one may be related to the spin-flop transition of the Fe^3+^ moment, while the higher one may be caused by the destruction of the AFM coupling between the FM components of the Dy^3+^ moment (induced by the magnetic field) and the Fe^3+^ moment (canted AFM) [29]. The magnetization behaviors measured in a pulsed higher magnetic field are shown in Figure 3d–f. Besides the low-field transitions, no additional transitions are induced up to 45 T, and the slopes of the *M*-*B* curves are almost constant, indicating that there is a strong AFM interaction in the Fe^3+^-sublattice [4].

In order to further investigate magnetic-field-induced transitions and the dynamic magnetization behavior of DyFeO_3_, the magnetization results were investigated by pulsed magnetic fields. In this work, the waveform of the pulsed magnetic field is shown in Figure 4a, which is a full-wave pulsed magnetic field (including four quadrants, Q_A_, Q_B_, Q_C_, and Q_D_) with a maximum field sweep rate of about 10^4^ T/s. When the magnetic field is along the *a*-axis and the temperature is below 4.2 K (see Figure 4b), in the field-increasing branch (quadrant Q_A_), the transitions resulting from the Dy^3+^-sublattice (marked with red arrows) and the field-induced spin flop in the Fe^3+^-sublattice (marked with black arrows) are observed. When the temperature ranges from 4.2 K to 50 K, only the transition, corresponding to the spin flop in the Fe^3+^-sublattice, could be observed, and its critical magnetic field decreases drastically with increasing temperature. Above 50 K, no obvious transition is observed (not shown). In the field-decreasing branch (quadrant Q_B_), the transition field moves to the low-magnetic-field region (lower than 1 T). In the field-increasing and field-decreasing branches of the negative magnetic field (quadrants Q_C_ and Q_D_), the magnetization behaviors are similar to those in Q_A_ and Q_B_. As shown in Figure 4c, the magnetization behaviors of the magnetic field along the *b*-axis are similar to those along the *a*-axis.

In Figure 4b,c, the temperature dependence of the transition field of the spin flop in the Fe^3+^-sublattice is different above and below 4.2 K. Below 4.2 K, the critical field (corresponding to the spin flop of the Fe^3+^ moment) is essentially unchanged with the temperature increasing. Above 4.2 K, the transition field (labeled with arrows) has a strong dependency on temperature. These results suggest the pinning of the Dy^3+^-sublattice on the spin flop of Fe^3+^ ions. With the temperature increasing, the pinning gradually becomes weaker due to the destruction of the long-range order of the Dy^3+^-sublattice. The transition field (labeled with arrows) of the Dy^3+^-sublattice has a strong temperature dependence, which indicates that the direction of the magnetic anisotropy (or anisotropy energy) of Dy^3+^ spins may also be disturbed by the magnetic field in the *ab* plane. The critical field moves toward the lower-magnetic-field region with the temperature increasing. To present the moving trend more clearly, the phase diagrams with the magnetic field parallel to the *a*- and *b*-axes are shown in Appendix A, respectively. When the magnetic field is applied along the *c*-axis, only the transitions related to the spin flop of Fe^3+^ ions are observed (see Figure 4d). Since the Dy^3+^ ion has strong Ising behavior and is localized in the *ab* plane, no obvious transition related to the Dy^3+^-sublattice was observed for lower magnetic fields. The critical field (corresponding to the spin-flop of the Dy^3+^ moment) moves toward the lower-magnetic-field region with the temperature increasing, and the temperature and magnetic field dependence of the critical behaviors are shown in Appendix A.

In the measurement of the electric polarization, Δ*P_c_* is a relative value, and it shows the change in *P_c_* induced by magnetic fields, and the applied magnetic field includes four quadrants: Q_A_, Q_B_, Q_C_, and Q_D_. As shown in Figure 5a, both the magnetic field and the electric field are parallel to the *c*-axis; with the magnetic field increasing (quadrant Q_A_), Δ*P_c_* jumps (labeled with red pentacles) are observed at *B_P_*(Fe) and a temperature below *T_N_*(Dy). At 2 K, the transition is observed in *B_P_*(Fe) ~3 T (labeled with a red pentacle), and the critical magnetic field is coincident with the result of magnetization measurement in the field-increasing branch (quadrant Q_A_). As the temperature increases, the critical field *B_P_*(Fe) moves to the low-field region, and a similar temperature dependence of the transition field is also observed in the magnetization curves (see Figure 4d). In the field-decreasing branch (quadrant Q_B_), Δ*P_c_* becomes zero at *B*’*_P_*(Fe) (marked with black triangles). In quadrant Q_C_ (the field-increasing branch of the negative magnetic field), the transitions are also observed at −*B_P_*(Fe) (marked with black diamonds). In the field-decreasing branches of the negative-magnetic-field region (quadrant Q_D_), a transition is observed at −*B*’*_P_*(Fe) (marked with crosses). Particularly, a metastable state (indicated by solid circles) and Δ*P_c_* reversal (marked with a cross) are observed around 3.1 K in the negative-magnetic-field region. The transitions are affected by magnetic fields and temperatures, which may originate from the complicated interactions between the anisotropy energy of the Dy^3+^-sublattice, the coupling energy between the Dy^3+^ and Fe^3+^-sublattices, and Zeeman energy.

According to the exchange striction model, *P_c_* is related to the spin flop of both the Fe^3+^-sublattice and the Dy^3+^-sublattice [18]; to reverse the *P_c_*, it is necessary to change the phase (0 or π) of the magnetic vector of either the Fe^3+^ or the Dy^3+^ ions. The magnetic vector of the Fe^3+^ ions is directly connected to the direction of the wFM of the Fe^3+^-sublattice. Thus, the field-induced Δ*P_c_* is observed when a large magnetic field is antiparallel to the *c*-axis. With the magnetic field decreasing and the temperature increasing, the interaction between the Dy^3+^ and Fe^3+^ ions, as well as the Zeeman energy, becomes weaker, and the magnetic anisotropy energies of the Dy^3+^-sublattices gradually dominate, which leads to the Δ*P_c_* reversal and metastable polarization states in the negative field (quadrants Q_C_ and Q_D_ of Figure 5a). On the other hand, the strong magnetic anisotropy of the Dy^3+^ ions becomes dominant, which drives the Dy^3+^ spins to its easy axis and leads to the change in the exchange striction and Δ*P_c_* reversal to a lower value. For the observed metastable state (marked with solid circles in Figure 5a), we assume that this is due to the spin-pinning effect of Dy^3+^ ions on the change in the wFM of the Fe^3+^-sublattice. The fact that metastable polarization behavior is more obvious when the temperature approaches *T*_N_(Dy) indicates that magnetic anisotropic Dy^3+^ is more easily magnetized by the magnetic field when the temperature approaches *T*_N_(Dy) than at lower temperatures.

For the pulsed magnetic field within the *ab* plane (as shown in Figure 5b–d), the change in the Dy^3+^ spins induced by the magnetic field also causes the change in exchange striction. The Δ*P_c_*-*B* curves with magnetic fields along the *a*- and *b*-axes were measured (where the electrical polarization is along the *c*-axis). As shown in Figure 5b (the magnetic field along the *a*-axis) and Figure 5c (the magnetic field along the *b*-axis), the sign of Δ*P_c_* is unchanged. This is totally different from the case with a magnetic field along the *c*-axis. The unchanged Δ*P_c_* suggests the simultaneous flop of both the Fe^3+^ and Dy^3+^ spins when the magnetic field (in the *ab* plane) reverses. At 2 K, with the magnetic field along the *a*-axis and *b*-axis and the field increasing (quadrant Q_A_), the field-induced Δ*P_c_* are observed at *B_P_*(Dy) ~0.8 T (*a*-axis, marked with a blue pentacle) and 0.5 T (*b*-axis, marked with red triangles), respectively. Both the critical fields are lower than that along the *c*-axis, and with the increasing temperature, the transition fields move further to the lower magnetic fields. The effects of the temperature and magnetic field on the critical behaviors are shown in Appendix A.

In DyFeO_3_, the AFM interaction in the Dy^3+^-sublattice is weak and mainly localized within the *ab* plane. The lower magnetic field in the *ab* plane will disturb the direction of the magnetic anisotropy (or the anisotropy energy) of Dy^3+^ ions; that is, the Dy^3+^ moments are easily magnetized by the magnetic field, resulting in a higher magnetic field sensitivity. At zero fields, the magnetic vector (the Ising axis) of the Dy^3+^ ion deviates by about 33° from the *b*-axis (as shown in the inset of Figure 3a), which leads to the different critical field of the field-induced Δ*P_c_* between the magnetic field along the *a*- and *b*-axes (see Figure 5b,c). In order to confirm the intrinsic effect of the magnetic field on polarization, the Δ*P_c_* was also investigated with zero electric fields (see Figure 5d), and a weaker change of the electrical polarization was observed in the d*P_c_*/d*B*-*B* curves, where the critical fields (marked with purple triangles) are coincident with those observed in the Δ*P_c_*–*B* curves (see Figure 5c). With the temperature increasing, the transition peaks of the Δ*P_c_* shift to the lower-field region. These experimental results indicate that the Δ*P_c_* is an intrinsic behavior and can be induced by the magnetic field alone.

## 4. Conclusions

In this work, the combination of anisotropy and exchange striction in DyFeO_3_ provides a guiding principle for designing high-sensitivity spin-driven multiferroicity. Especially in the *ab* plane, the direction of the magnetic anisotropy (or the anisotropy energy) of the Dy^3+^ ions can be modulated by a smaller magnetic field, which alters the exchange striction and leads to a Δ*P*_c_ sensitive to the external magnetic field. That is, the combination of the magnetic anisotropy of the Dy^3+^ spin and the exchange striction between the Fe^3+^ and Dy^3+^ spins leads to the Δ*P**_c_*. The Δ*P**_c_* exhibits high magnetic-field sensitivity. This work deepens the understanding of the effects of the multiple magnetic orders on the magnetoelectric coupling of multiferroic DyFeO_3_ and will be beneficial in searching for novel multiferroic material systems with high magnetic-field sensitivity.

## Figures and Tables

**Figure 1 nanomaterials-12-03092-f001:**
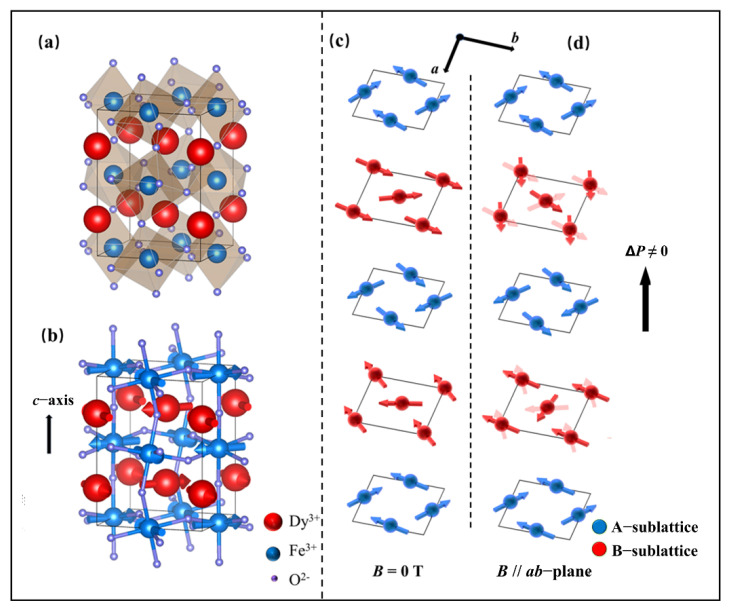
(**a**) The crystal structure and (**b**) magnetic configuration of DyFeO_3_ below the Dy^3+^ ordering temperature. Two AFM sublattices nesting with each other at (**c**) zero field and (**d**) applied magnetic field.

**Figure 2 nanomaterials-12-03092-f002:**
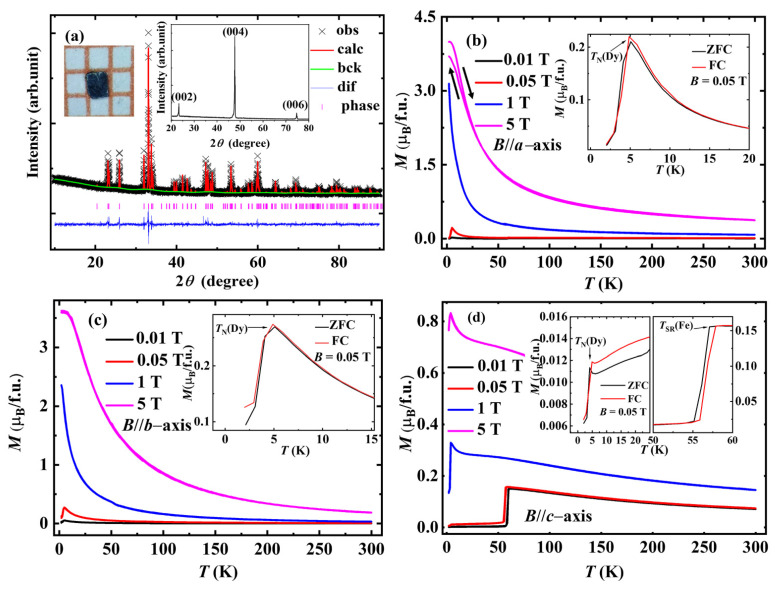
(**a**) The XRD patterns of DyFeO_3_ powder and single crystal (right inset) and the DyFeO_3_ single-crystal sample morphology (left inset). (**b**–**d**) The temperature dependence of the magnetization of the DyFeO_3_ single crystal with magnetic fields along the *a*-, *b*-, and *c*-axes, respectively. The partial magnification near the transition temperatures of ZFC and FC curves measured at 0.05 T are shown in the corresponding insets.

**Figure 3 nanomaterials-12-03092-f003:**
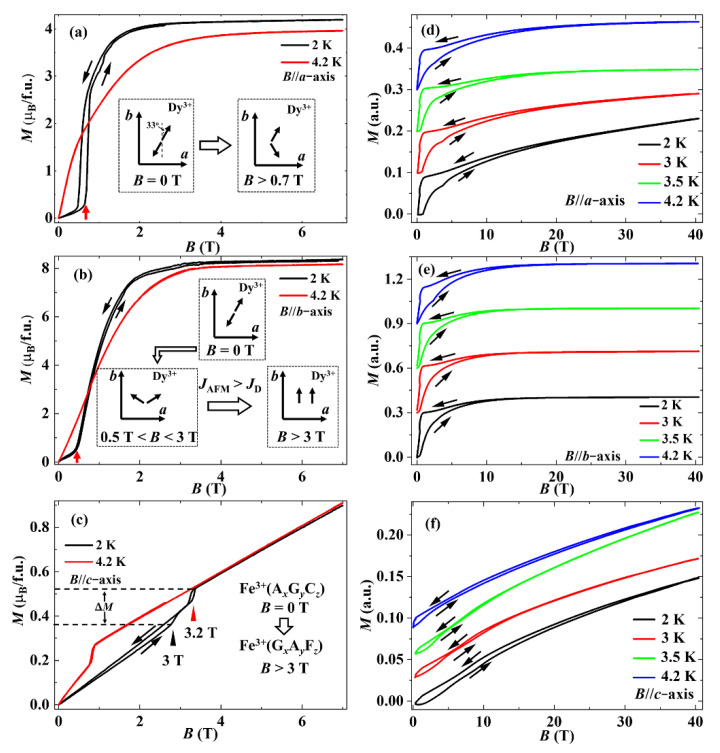
(**a**–**c**) Magnetization as a function of the static magnetic field along *a*-, *b*-, and *c*-axes, respectively. (**d**–**f**) The magnetization curves measured along the *a*-, *b*-, and *c*-axes under pulsed high magnetic field, respectively. The curves in the (**b**,**e**,**f**) are offset for clarity. *J*_AFM_ and *J*_D_ are the AFM interaction strength and anisotropy energy of the Dy^3+^-sublattice, respectively.

**Figure 4 nanomaterials-12-03092-f004:**
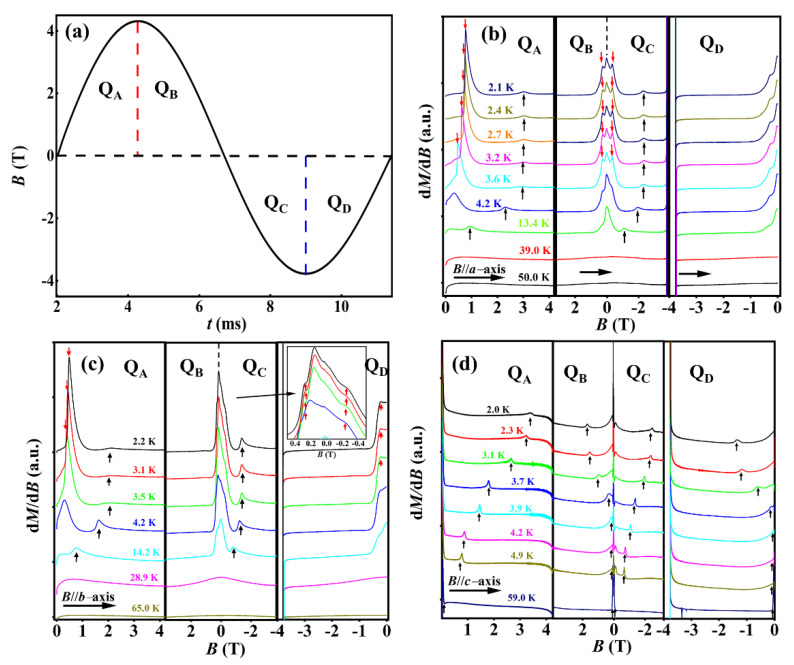
(**a**) The waveform of the pulsed magnetic field. (**b**–**d**) Magnetization derivative (d*M*/d*B*) as a function of the magnetic field measured along the three principal axes. The inset in (**c**) is the enhancement of the d*M*/d*B* around the zero-magnetic-field regions. The curves of (**b**–**d**) are offset for clarity. Q_A_, Q_B_, Q_C_, and Q_D_ are the four quadrants of the pulsed magnetic field; all the transitions are labeled with arrows.

**Figure 5 nanomaterials-12-03092-f005:**
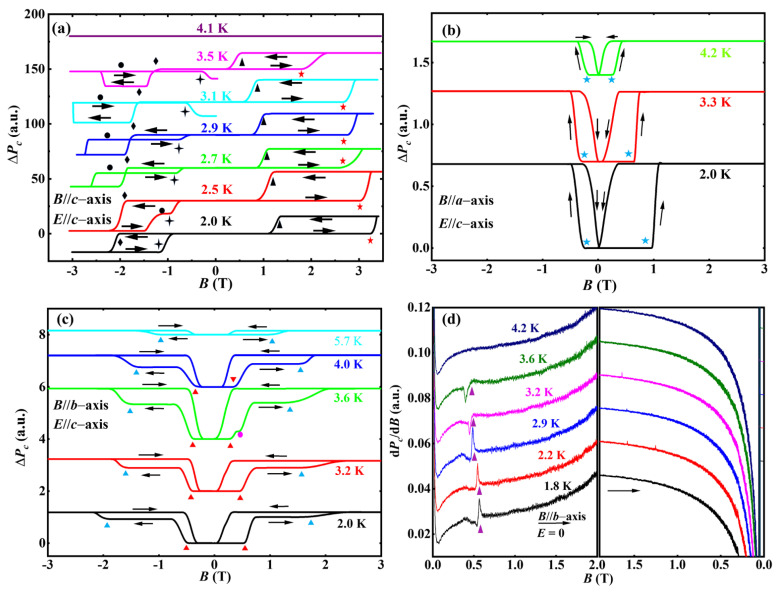
(**a**–**c**) Electric polarization as a function of pulsed magnetic fields, measured under the pulsed magnetic field along *c*-axis (**a**), *a*-axis (**b**), and *b*-axis (**c**), where the electric fields (*E* = 1.5 kV/cm) are along *c*-axis. (**d**) The magnetic field dependence of d*P*_c_/d*B* measured at various temperatures with the applied pulsed magnetic field along *b*-axis and *E* = 0. The curves are offset for clarity. The various symbols of red pentacle, black triangle, black diamond, cross, solid circle, blue pentacle, red triangle, and purple triangle represent the transitions in the curves. The sweep directions of the magnetic field are labeled by black arrows.

## Data Availability

Not applicable.

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
