# Peer review of "High-Magnetic-Sensitivity Magnetoelectric Coupling Origins in a Combination of Anisotropy and Exchange Striction"

_nanomaterials, 2022, doi:10.3390/nano12183092_

Round 1
Reviewer 1 Report (New Reviewer)
The paper “High magnetic sensitivity magnetoelectric coupling origins in a combination of anisotropy and exchange striction” by Zhuo Zeng et al. deals with actual problem of the magnetoelectric effect and novel multiferroic materials. Authors report on experimental study of magnetoelectric effect in DyFeO3 single crystal in magnetic field with varied magnitude applied in different geometries. The dependence of ME effect on the direction and magnitude of magnetic field has been shown, the measurements confirming the appearance of ferroelectric polarization have been performed. Authors presented the detailed analyzed the observed behavior and anomalies of magnetization and polarization. The subject is prospect and interesting, and I recommend the manuscript to be published in the Nanomaterials journal.
However, I have several comments
11. The first one concerns the terminology, authors refer to DyFeO3 single crystal as novel high magnetic-sensitivity single-phase multiferroic materials, but DyFeO3 is known by its multiferroic properties at least since 2008 when magnetoelectric effect in DyFeO3 has been observed experimentally [Tokunaga, Y.; Iguchi, S.; Arima, T.; and Tokura, Y. (Ref.4) ]and predicted theoretically [Zvezdin, A. K., & Mukhin, A. A. (2008). Magnetoelectric interactions and phase transitions in a new class of multiferroics with improper electric polarization. JETP letters, 88(8), 505-510]. Since 2008, a number of researchers have undertaken their efforts to study magnetoelectricity in rare earth orthoferrites and related orthochromites.
22. I suggest to estimate the value of magnetoelectric (ME) coefficient (ME tensor components) for the cases magnetic field applied in – plane (ab – plane) and out – of plane (along c – axis), this magnitude is important for the analysis of ME properties of multiferroic materials, it also allows to make the comparisons with previously reported results (e.g. Ref.4).
33. I would like to recommend to the authors theoretical and experimental works on the analysis of the mechanisms of the ME effect in perovskite multiferroics, including rare-earth orthoferrites (RFeO3, in particular DyFeO3), e.g. [1] Zvezdin, A. K., & Mukhin, A. A. (2008). Magnetoelectric interactions and phase transitions in a new class of multiferroics with improper electric polarization. JETP letters, 88(8), 505-510, [2] Bousquet, E.; Cano, A. Non-collinear magnetism in multiferroic perovskites. J. Phys. Condens. Matter 2016, 28, 123001, [3] Zvezdin, A. K., Z. V. Gareeva, and X. M. Chen. "Multiferroic order parameters in rhombic antiferromagnets RCrO3." Journal of Physics: Condensed Matter 33.38 (2021): 385801, [4] Rajeswaran B, Sanyal D, Chakrabarti M, Sundarayya Y, Sundaresan A and Rao C N R 2013 Europhys. Lett. 101 17001,
there they can find the analysis of ME mechanisms, phase transitions in orthoferrites, expected dependences of polarization behavior in magnetic fields of different geometry
Author Response
Please see the attachment

Reviewer 2 Report (New Reviewer)
The manuscript by Zeng et al. discusses the possibility of novel high magnetic sensitivity single-phase multiferroic in the form of DyFeO3 single crystals with the proposal of two nesting AFM sublattices. The manuscript has a few deficiencies following which it may be reconsidered for publication. They are listed below.
1. The use of english language in the manuscript leaves a lot to be desired. The third sentence in the abstract makes little sense as phrased and very similar sentences can be found through the rest of the manuscript. In some places, the scientific argument becomes difficult to interpret. Strong revision with due diligence is recommended.
2. The authors quote prior works on DyFeO3 and then say that 'we propose a novel ME system with two nesting sublattices' ! Is this just a re-interpretation of the magnetic structure for DyFeO3 and if yes, the claims should be phrased differently. Also, it is interesting that the claims in Figures 1a and 1b have not been justified with any theory or simulations but just made. While the claims may be reasonable, the reader does not understand why these claims have been made out of thin air at this stage other than being told so. I think this part of the manuscript needs further work in terms of justifying the motivation and the experimental case for the existence of two nested magnetic sublattices at all. Just stating that there may be a resemblance of the structures (proposed vs DyFeO3) is insufficient in my humble opinion.
3. The polarisation generated in in Figure 5 at 2K is exceptionally small when compared with typical polarisation observed in ferroelectrics (~0.005 uC/cm^2). Can the authors comment on this magnetic-field induced electric polarization and how its magnitude compares with other single phase multiferroics illustrating spin-driven ferroelectricity. The more subtle question is why such small polarisation is of significant value or are there possibilities for tuning the response to achieve more useful values?
4. Several of the statements interpreting the magnetic field dependence behaviour remain not well qualified. An example among many is 'there are complicated interactions between the anisotropy energy of the Dy3+-sublattice, the coupling energy between the Dy3+ and Fe3+ sublattices, and Zeeman energy.' While true, what is the clear basis for making such statements and how much confidence can the reader have in the speculative arguments when the data has not been backed up with solid arguments? While the exchange striction model seems plausible, there is not evidence provided to believe that it is the dominant mechanism. It is necessary that the authors qualify such statements with due care and the onus of correctness and explanations sits with them.
Round 2
Reviewer 2 Report (New Reviewer)
The authors have made adequate changes to the manuscript for it to be acceptable for publication.
Author Response
Please see the attachment

This manuscript is a resubmission of an earlier submission. The following is a list of the peer review reports and author responses from that submission.
Round 1
Reviewer 1 Report
The authors have put together a thorough study describing their data and theoretical explanation of high ME effect in DFO single crystals. It is currently of high community interest the nature of the recently perceived antiferromagnetic coupling in the RE metals such as Dy and how it can be tuned in multi-sublattices to produce novel systems. Here the authors describe such a system in DFO and postulate their effects are due to exchange striction between the Fe and Dy sublattices. While the e-field induced and piezoelectric ME effects shown here are at low temperature, the high degree of control and sensitivity is of high interest to the community and could help in designing practical systems at room temperature.
The article is well written and the experimental methods are sound. Measurements of P-V loops and ME effect are undertaken with the accepted methods (silver past electrical contacts on either side of the sample).
Electric polarization as a function of B field diagrams in Fig 5 are clear and concise to the reader. Measurements were taken in pulsed fields and this has been made clear in the text.
I therefore recommend that this article is published without further delay in Nanomaterials.
I recommend only moderate checking of the use of articles for English grammar editing.
Reviewer 2 Report
The manuscript "High magnetic sensitivity magnetoelectric coupling origins in a combination of anisotropy and exchange striction" by Zhuo Zeng and co-workers describes magnetic field dependent measurements of the ortheferrite DyFeO3 as a single crystal. Special is that the authors are able to conduct measurements in magnetic fields up to 50T. Further, the authors claim " that the combination of multi-magnetic order parameter modulation, gives rise to a high magnetic sensitivity ME coupling in single-phase materials".
Overall, the manuscript contains some interesting data. Looking at the presentation, how data are described and interpreted and explained, the manuscript is nowhere near to be fit to be acceptable in any journal.
Why do I come to this conclusion. First, I have some issues with the langage. On the one hand, the English is quite good. On the other hand descriptions given are in a number of places either not correct the way they are phrased, or they could be correct if the context would be different. But this is to some extend a matter of language. In places I also have reasonable doubt that the authors really understand some of the physical concepts since they handle the wording wrongly. For example in line 45/46. The content of the sentence as it reads is wrong in a couple of ways. If the authors would have said that if a magn. field is applied along the c-axis at 4K or below the Dy ordering temperature, than the sentence would have been partly correct because the symmetry will indeed be again GxAyFz up to 50K. The Morin transition however, will not be induced by the magnetic field. The Morin transition is also not a phase transition as written several times in the manuscript. It is, as the more general terms expresses, a reoriention of ordered spins and this reoriention is triggered by the Dy-Fe interaction. The spin reorientation for DyFeO3 is special in a sense that the transition width is sharp unlike most other orthoferrites.
The electrical polarization is a vector. This means P usually has a distinged direction. For DFO it is supposed to be
the c-direction with the magnetic field parallel to c. The referee is not aware of reports saying anything different for bulk DFO. Though, what is the bases for the statement in line 56/57 about ME coupling? Ref 4 clearly shows P||c. If you later claim to have ME coupling in different crystalline directions, I would argue that you have twins in your crystal, or other orientations and you measure all of it. Can you clarify this? You only show the out-of-plane x-ray scan and claim high quality single crystals. How about in-plane?
A general comment on your DFO crystal, despite the fact that the lattice constants seemed to be pretty OK, the reference for structure one typically would go for is the ICSD data base. When you would have read the literature, you may have noticed that even for reported single crystal data, there are noticible differences in these papers and hence also noticible differences in properties. E.g. T_SR is ranging between 40 and 70K and it is supposedly the same material? Yours with 59K is more towards the upper end. Something clearly is inconsistent. Ref. 20 is reporting very different lattice constants and I am not certain if the data reported there can be generalized. Within their own context, the data are fine, beyond that this is not clear. Looking what I have seen from your data properties are different to what I know from the literature and what we have measured on our own crystals. This simply means for the moment, you show some data from your crystal which are similar to what is known, others deviate. Having said that, most graphs you show are very badly presented and for the simple things like transitions to estimate a temperature or a field from, is difficult or impossible do (see e.g. Fig. 2d inset). You also limit yourself this way because you come to wrong conclusions because you miss the obvious.
Line 112/113 ZFC/FC measurements, I disagree with the statement of the authors. Most of the time we do measure differences between ZFC and FC, also others I talked to or what is reported in the literature. If correct, this seems something very specific to your measurements.
Line 117, the statement on wFM of Dy, where do you deduce this from? You would be the first one to report wFM in DFO single crytals originating from Dy. In all likelyhood, your statement is wrong.
Line 119, there is no phase transition. This is one example where I think you have no understanding what the concept of a phase transition is.
Line 125/126, how can the magnetic field destroy the magnetic anisotropy? Please explain?
Line 131, the reference to the Fig. S1, yes, you can find it there. But no explanations are given what am I looking at and how to understand all the different data points. This is not acceptable. The reader is supposed to figure out what the authors mean?
Lines 153-156: If I understand Fig. 3 correctly, you did the M(H) at 2 and 4.2K, correct? This means, you have an AFM ordered Dy spin lattice. A FM component arisses only if your spins are FM ordered, or if you have a tilting of the AFM lattice. With Dy in DFO, it should be the latter according to what you write and you take your knowledge from ref. 26. However, from symmetry you will not have a canting of the Dy spin lattice. Hence, where does your wFM from the Dy comes from you are claiming to have? With the Dy-wFM in ref. 26, this was never really clear to me since in the temperature range they measured Dy is still the paramagnet, and the Fe spin lattice is really ridgid. Their experiments would mean, that they order the Dy spins with the applied field. When you analyize M(H)-loops taken at different temperatures, you will notice, that they are down to 30K largely paramagnetic in character. Though, this is somewhat of a discrepency. Since your M(H)s are looking also very different, what do you have?
Figure 3d-f, why can you not convert the measured magnetization into the same units like you have it for the SQUID-based measurements? Your way of plotting (without explanations) makes it impossible to compare the low field and the high-field data.
Your explanations for Fig. 4 are really badly done. Likewise the plotting is inconsistent. You change the field between -4 to 4T, OK, but in Fig. 4b-d, you plot only sections of it and those you plot in parts asymmetric in terms of field. It is very difficult to follow what you are trying to say and I understood not much. Likewise everything related to Fig. 5. This is probably interesting, but difficult to follow or judge fairly what you would like to say. You don't even tell the reader if you are doing pyromeasurements or loops, or what is the meaning of the many accronyms. These parts require a restart. In their current state, it is difficult to know what you are on about.
Round 2
Reviewer 2 Report
The manuscript by Zheng et al on the "High magnetic sensitivity magnetoelectric coupling origins in a combination of anisotropy and exchange striction" has improved after the revision, but it clearly still contains errors or inaccuracies and is therefore not suitable for publication.
L32/33 the authors write about a wFM and FE state. Therefore "lower temperatures" means below the Dy ordering temperature. Below T_N,Dy, however, DyFeO3 is not a FM, neither a weak one nor a conventional one. The latter would be a contradiction since DFO has two AFM spin systems below 4.2K. I leave it to the authors to figure out what it is. And that the expression "wFM" is wrong, we know very well from personal experience.
L41, there is no "the wFM", it is a wFM
L42, you quote for T_SR your measured value, but there is a range of different values in the literature. Since this is the introduction you give the reader a very wrong impression of what T_SR can be.
L46/47 this statement is still wrong. DFO will not become FE when you apply a field below T_SR however hard you try. And I don't think, this is acctually stated in ref [20].
In the next paragraph, you are on about ME coupling. Your thinking and than hence writing about ME is incorrect. The coupling mechanism between the electric and magnetic dipol moment is not the origin of FE in DFO or in any other MF material. It is also known what is the origin of FE in DyFeO3 is, and this has very little to do with an anisotropy of Dy3+. You even write correctly at another place how DFO becomes a polar material. Though, I do not really understand the idea you outline in the paragraph starting at line 66 with the two magnetic systems. What the magnetic soft and hard axis is, is still given by the Fe spin system to a large extend.
Your insets in Figure 2d, you measure a temperature hysteresis for the SR? If the SR would be a phase transition which it is not, maybe but than the order is wrong. You have an issue with the temperature control of your measurements. Also, what does the peak at low temperatures (prob. around 5K) in the moment vs T at low temperature mean?
L 123/124 wrong statement: the Morin transition is not induced by a magnetic field
L151, you are misquoting reference 24. Ref. 24 is quoting Phys. Rev. B 93, 140403(R) (2016) when they summarize what is written in the literature. And honestly, I don't think PRB 93 is correct.
L154, what do you mean by that the Dy3+-sublattice was almost polarized by the magnetic field? I don't understand the terminology.
L163 you are suggesting that the applied magn. field is destroying the interaction between Fe and Dy? Between the two spin lattices, you can probably change the interaction, but destroy the coupling? Please remember, in terms of Zeeman engery, a magnetic field of 10T corresponds to 1K. This is not much when compared to typical engery scales. I don't think your assumption is correct.
Polarization measurements, it is nice that you tell me how you do it. But you still don't write it in the manuscript where it should be. This is essential information, but instead you write it in some detail in the supplement. This is fine, but the supplement is not the manuscript. It is an add-on with additional information related to the manuscript. You use the SI like it is part of the manuscript which it should not be.
ME coupling: These measurements do not show a mutual dependence between P and M. You have shown that the electrical polarization is magnetic field induced and the FE phase in a crystal above 5K cannot be induced by a magnetic. And you measured the magnetic field dependence for E||c. How about E||a or b? You do not mention this.
